# Preliminary Study: Comparison of Antioxidant Activity of Cannabidiol (CBD) and α-Tocopherol Added to Refined Olive and Sunflower Oils

**DOI:** 10.3390/molecules24193485

**Published:** 2019-09-26

**Authors:** Matilde Tura, Mara Mandrioli, Tullia Gallina Toschi

**Affiliations:** Department of Agricultural and Food Sciences, Alma Mater Studiorum-University of Bologna, Viale Fanin 40, 40127 Bologna, Italy; matilde.tura2@unibo.it (M.T.); mara.mandrioli@unibo.it (M.M.)

**Keywords:** cannabidiol (CBD), lipid oxidation, α-tocopherol, antioxidant activity, oxidative stability, free radicals

## Abstract

This study evaluates the antioxidant activity of cannabidiol (CBD), added to model systems of refined olive (ROO) and sunflower (SO) oils, by measuring the peroxide value, oxidative stability index (OSI), electron spin resonance (ESR) forced oxidation, and DPPH^•^ assays. Free acidity, a parameter of hydrolytic rancidity, was also examined. CBD was compared using the same analytical scheme with α-tocopherol. CBD, compared to α-tocopherol, showed a higher scavenging capacity, measured by DPPH^•^ assay, but not better oxidative stability (OSI) of the oily systems considered. In particular, α-tocopherol (0.5%) showed an antioxidant activity only in SO, registered by an increase of more than 30% of the OSI (from 4.15 ± 0.07 to 6.28 ± 0.11 h). By ESR-forced oxidation assay, the concentration of free radicals (μM) in ROO decreased from 83.33 ± 4.56 to 11.23 ± 0.28 and in SO from 19.21 ± 1.39 to 6.90 ± 0.53 by adding 0.5% α-tocopherol. On the contrary, the addition of 0.5% CBD caused a worsening of the oxidative stability of ROO (from 23.58 ± 0.32 to 17.28 ± 0.18 h) and SO (from 4.93 ± 0.04 to 3.98 ± 0.04 h). Furthermore, 0.5% of CBD did not lower dramatically the concentration of free radicals (μM) as for α-tocopherol, which passed from 76.94 ± 9.04 to 72.25 ± 4.13 in ROO and from 17.91 ± 0.95 to 16.84 ± 0.25 in SO.

## 1. Introduction

Cannabidiol (CBD) is a non-psychoactive cannabinoid present in *Cannabis sativa* L., and unlike tetrahydrocannabinol (THC), CBD has a very low affinity for the cannabinoid CB1 and CB2 receptors [1]. From the chemical structure of CBD (Figure 1), it is easy to recognize the presence of two hydroxyl groups that can endow it with antioxidant activity [2]. CBD is a cyclohexene that is substituted in position 1 by a methyl group, in position 3 by a 2,6-dihydroxy-4-pentylphenyl group, and in position 4 with a prop-1-en-2-yl group [3]. According to Borges et al. 2013, CBD has potential antioxidant activity because of the fact that the cation free radicals show several resonance structures in which the unpaired electrons are mainly distributed on the ether and alkyl groups, as well as on the benzene ring [4].

One of the most common antioxidants is α-tocopherol, which is a key component in biological systems as it integrates into the cell membranes to protect their constituents [5].

Tocopherols are natural antioxidants and liposoluble vitamins with antioxidant action both in vivo and in vitro. They can be classified as four derivatives [6], depending on the position and the number of methyl groups on the chromanol ring [7], called alpha (Figure 1), beta, gamma, and delta tocopherol. These tocopherol isomers differ in their antioxidant activities, with the highest antioxidant activity found for α-tocopherol [6]. In fact, when α-tocopherol becomes a free radical the resulting α-tocopheroxyl species is able to delocalize the free electron, thus forming a more stable and less reactive intermediate [5].

The human body cannot synthetize tocopherols, hence they must be included in the diet [6]. According to Muhammad et al. 2012, the amount of total tocopherol present in crude sunflower oil is between 447 and 900 mg/g of oil, with extreme values between 389 and 1873 mg/g of oil. Up to 90% of the total content of tocopherols in sunflower oil is represented by α-tocopherol [5]. If sunflower oil is refined, the total content of these compounds is lower, and in fact, for vegetable oils, up to 32% of native tocopherols are removed during the refining process [7]. Regarding the content of α-tocopherol in refined olive oil, Schwartz et al. 2008 [8] reported a content of α-tocopherol equivalents equal to 17 mg/100 g.

Several assays can be used to evaluate the in vitro antioxidant capacity of extracts, such as lipid peroxidation inhibition assay, ferrous-ion chelating activity, inhibition of DPPH^•^(2,2-diphenyl-1-picrylhydrazyl), inhibition of 2,2′-azino-*bis*(3-ethylbenzothiazoline-6- sulphonic acid)—ABTS•+ [9], superoxide dismutase mimetic activity, and many others. All these assays have their pros and cons, but when the antioxidant activity is evaluated it is essential to consider that these methods have specificity in relation to the mechanism of action, target, pH, time, and temperature; furthermore, different standards are used to construct the calibration curves that produce quantitative results in terms of antioxidant activity. Therefore, no single in vitro “antioxidant activity” assay will reflect the total antioxidant capacity [10,11], but all possible measures have a partial meaning that is strictly related to the method of measurement and the setting (e.g., biological, food system, pure oil). Moreover, antioxidants may respond in a different way to various radical or oxidant sources [12]. In fact, the activity of antioxidants depends not only on their structural features, but also on many other factors, such as concentration, temperature, type of substrate, and physical state of the system, as well as on the numerous micro-components acting as pro-oxidants or synergists [12].

Free radicals act in both food and biological systems; for this reason, dietary antioxidants play an important role in controlling an excess of free radicals [5].

In food systems, free radicals lead to oxidative rancidity whereby polyunsaturated fats break down, causing the formation of off-flavors [5,13].

By leading to the formation of off-flavors, the oxidative process not only makes a product less attractive, but also results in the formation of toxic products, such as oxidized polymers, because of the destruction of fatty acids [14]; for this reason, oxidation is a major problem that affects edible oils [15]. Fats and oils are susceptible to oxidation in the presence of catalytic systems such as light, heat, enzymes, metals, metalloproteins, and microorganisms. These catalysts lead to complex processes of oxidation, and in particular autoxidation, photooxidation, and thermal or enzymatic oxidation, most of which involve free radicals and/or other reactive species as the intermediate [16]. Oxidative stability is defined as the resistance to oxidation during processing and storage of oils and is very important to determine the quality of an oil and its shelf life. In fact, it is the period necessary to reach the critical point that determines the sensory changes or a sudden acceleration of the oxidative process [14].

The aim of the present study is to evaluate the antioxidant capacity of cannabidiol (CBD), added to two lipid matrixes, namely refined olive oil (ROO) and refined sunflower oil (SO). Herein, the antioxidant capacity of cannabidiol was compared with the same oils to which α-tocopherol (AT) was added.

## 2. Results

### 2.1. Determination of CBD Content

The determination of CBD was conducted in the oily solutions produced, while, as well-known, refined olive oil and sunflower oil do not present any CBD content. The analysis was carried out using the HPLC-UV method (described in Section 4.2) and the amounts of CBD present in the oily solutions were consistent with those added to the oils. The recoveries were calculated with respect to the three concentrations investigated (0.01%, 0.1% and 0.5%), by applying the following formula:R (%) = [(Cf − C)/Cc] × 100,(1) where: Cf is the amount of cannabinoid in the sample added to the analyte under examination. C is the quantity of cannabinoid determined in the sample not added. Cc is the quantity of analyte added to the sample.

The results were between 96.42% and 125.82%, with a mean value of 110.15%; for this reason, CBD can be considered completely solubilized.

### 2.2. Determination of α-Tocopherol Content

The starting concentrations of α-tocopherol were calculated in refined olive oil and in sunflower oil, which were used to prepare solutions with different concentrations of α-tocopherol. The starting concentration of this compound in refined olive oil was 0.018% and in sunflower oil was 0.061%. The total quantities of α-tocopherol found in the oils were consistent with the quantities of alpha tocopherol added to the oil also considering that they are naturally present. Even in this case, as described above, the yield was calculated. Regarding recovery of α-tocopherol added, the mean value was 93.36%.

### 2.3. Peroxide Value

Peroxide value is one of the most frequently used quality parameters, as it measures the amount of total peroxides which are considered as primary oxidation products [17]. The samples of refined olive oil and sunflower oil used for the preparation of the oils added with the two active ingredients were the same. Given the instability of the peroxides and, as a consequence, the rapid variation of the peroxides value over time, the evaluation of this parameter was carried out on refined olive and sunflower oils whenever samples with CBD or α-tocopherol added were prepared. Therefore, although there is a change in the peroxide value during the storage period (two months in the dark at 4 °C), it was possible to compare the antioxidant activity of the sample with the active ingredient (CBD or α-tocopherol) added and the related sample of oil in which the active ingredient was not added.

The initial peroxide values for refined olive oil and sunflower oil, used to prepare solutions with the three different concentrations of CBD (0.01%, 0.1%, and 0.5%), were 1.93 meq O_2_/kg of oil and 5.00 meq O_2_/kg of oil, respectively (Table 1).

Refined olive oil with CBD added at the three concentrations showed lower values than the sunflower oil with CBD at the same concentrations. In the first case, they were between 2.47 meq O_2_/kg of oil and 2.99 meq O_2_/kg of oil, while in the second case they were between 6.04 meq O_2_/kg of oil and 14.73 meq O_2_/kg of oil (Table 1).

The initial peroxide values for refined olive and sunflower oils, used to prepare solutions with three different concentrations of α-tocopherol (0.01%, 0.1%, and 0.5%) were 0.15 meq O_2_/kg of oil and 12.31 meq O_2_/kg of oil, respectively (Table 2). Refined olive oil with α-tocopherol added at the three different concentrations showed lower values than the sunflower oil added with α-tocopherol at the same concentrations; in particular, it was between 0.52 meq O_2_/kg of oil and 0.62 meq O_2_/kg of oil for refined olive oil-based samples and between 13.43 meq O_2_/kg of oil and 15.08 meq O_2_/kg of oil for sunflower oil-based samples (Table 2).

### 2.4. Free Acidity

The free acidity value is related to the degree of lipolysis of triglycerides and is a quality control parameter that needs to be assessed after the production and during shelf-life [18]. The initial free acidity value found for refined olive oil and sunflower oil, used to prepare solutions with the three concentrations of CBD (0.01%, 0.1%, and 0.5%) was 0.15 mg KOH/g of oil and 0.11 mg KOH/g of oil, respectively (Table 1).

Refined olive oil added with CBD at the three concentrations showed higher values than sunflower oil with CBD at the same concentrations; in the first case they were between 0.18 mg KOH/g and 0.27 mg KOH/g, while in the second case they were between 0.06 mg KOH/g and 0.08 mg KOH/g (Table 1).

The initial free acidity value found for refined olive oil and sunflower oil, used to prepare solutions with the three concentrations of α-tocopherol (0.01%, 0.1%, and 0.5%) was 0.17 mg KOH/g of oil and 0.20 mg KOH/g of oil, respectively (Table 2).

Refined olive oil and sunflower oil with α-tocopherol added at the three concentrations showed similar values; in the first case they were between 0.17 mg KOH/g and 0.21 mg KOH/g, while in the second case they were between 0.17 mg KOH/g and 0.21 mg KOH/g (Table 2).

### 2.5. Oxidative Stability Index (OSI)

OSI measures the induction period by plotting conductivity against time [19]. The initial OSI- time values found for refined olive oil and sunflower oil, used to prepare solutions with the three concentrations of CBD (0.01%, 0.1%, and 0.5%) were 23.58 h and 4.93 h, respectively (Table 1).

Refined olive oil with CBD added at the three concentrations showed higher values than the sunflower oil with CBD added at the same concentrations; in the first case they were between 17.28 h and 23.33 h, while in the second case they were between 3.98 h and 4.90 h (Table 1).

The initial OSI-time value found for refined olive and sunflower oils, used to prepare solutions with the three concentrations of α-tocopherol (0.01%, 0.1%, and 0.5%) was 28.35 h and 4.15 h, respectively (Table 2).

Refined olive oil with α-tocopherol added at the 3 concentrations showed higher values than the sunflower oil with α-tocopherol added at the same concentrations; in the first case they were between 24.88 h and 28.23 h, while in the second case they were between 4.20 h and 6.28 h (Table 2).

### 2.6. Electron Spin Resonance

Lipid oxidation is a free radical chain reaction, and for this reason electron spin resonance (ESR) is a valuable tool for detection and quantification of lipid-free radicals. In fact, radicals with unpaired electron have unique magnetic properties [20]. Studies concerning radicals deriving from lipid oxidation, which are very unstable, are carried out using the spin trap technique and the most used spin trap is *n*-tert-butyl-α-phenylnitrone (PBN). PBN acts as a trap for temporary radicals and as their scavenger, which is why when there are both antioxidants and PBN in a solution, a competition mechanism occurs and the effect of the interaction among PBN, lipid radicals, and antioxidants depends on the type of oil and antioxidant [21]. The stable radicals produced by PBN spin-trapping are nitroxides with a spectra characterized by three hyperfine lines for the coupling between the electron spin (S = 1/2) and the nitrogen nuclear spin (I = 1) [22].

The results obtained with ESR analysis are expressed as concentration of free radicals (μM) after 240 min of oxidation forced assay (Table 3 and Table 4) at 110 °C. The analysis was carried out by applying a temperature of 110 °C in order to have the same temperature applied for the OSI analysis. The concentration of free radicals is quantified on the basis of a calibration curve, as shown in Section 4.7.

The results were also expressed as concentration of free radicals after 20 min of ESR-forced oxidation assay, since the first measurement still fell within the instrumental noise. The trends are similar to those found after 240 min (Table 3 and Table 4) with the exception of sunflower oil with 0.5% CBD, which showed a decrease in the concentration of free radicals after 20 min compared to sunflower oil alone; this marked decrease was not evident after 240 min (Table 3). Only the results after 240 min are commented upon in order to better highlight the trends observed.

The initial ESR values found for refined olive oil and sunflower oil, used to prepare solutions with the three concentrations of CBD (0.01%, 0.1%, and 0.5%) were 76.94 μM and 17.91 μM, respectively.

Refined olive oil with CBD added at the three concentrations showed higher values than the sunflower oil with CBD at the same concentrations. In the former they were between 70.98 μM and 83.85 μM, while in the latter they were between 16.84 μM and 25.37 μM (Table 3).

For refined olive oil with CBD added, the addition of 0.01% of this cannabinoid resulted in an increase in the concentration of free radicals compared to that obtained for the refined olive oil alone. In contrast, the concentration of free radicals decreased for samples of refined olive oil with 0.1 and 0.5% of CBD (Table 3).

For sunflower oil, the addition of 0.01% and 0.1% of CBD resulted in an increase in the concentration of free radicals compared to sunflower oil alone; the addition of 0.5% CBD led to a decrease in the concentration of free radicals (Table 3).

The initial ESR values for refined olive oil and sunflower oil used to prepare solutions with the three concentrations of α-tocopherol (0.01%, 0.1%, and 0.5%) were 83.30 μM and 19.21 μM, respectively (Table 4).

Refined olive oil with α-tocopherol showed higher values of free radicals than sunflower oil with α-tocopherol at the same concentrations; in the first case they were between 11.23 μM and 81.22 μM, while in the second they were between 6.90 μM and 22.61 μM (Table 4).

In refined olive oil with α-tocopherol, as the concentration of α-tocopherol increased the concentration of free radicals in the sample decreased.

The addition of 0.01% α-tocopherol to sunflower oil caused an increase in the concentration of free radicals, whereas at the other two concentrations the decrease in free radicals was directly proportional to the concentration of α-tocopherol added, compared to that for sunflower oil alone (Table 4).

### 2.7. DPPH^•^ Radical Scavenging Activity Assay

DPPH^•^ was used to evaluate the free radical scavenging effectiveness of antioxidants in different substances. Antioxidants are able to reduce the radical DPPH^•^ to the yellow-colored diphenyl-picrylhydrazine. The method is based on the reduction of DPPH^•^ in an alcoholic solution in the presence of a hydrogen-donating antioxidant because of the formation of the non-radical form DPPH-H in the reaction [23].

The samples analyzed showed some differences, and in particular in oils with α-tocopherol after 30 min. Although they had a lower absorbance than the initial measurement, they did not show a yellow color, but remained purplish. In contrast, samples with cannabidiol at a concentration of 0.5% after 30 min already showed a yellowish color (Figure 2).

The theoretical value of EC_50_ was calculated by taking into consideration the concentration of the oil, and the concentration of each sample was calculated based on the exact weight of the sample used to carry out the extraction.

The theoretical EC_50_ values for refined olive oil and sunflower oil, used to prepare solutions with the three concentrations of CBD, were 1.49 g/mL and 12.86 g/mL, respectively.

For refined olive oil, the addition of 0.01% CBD resulted in a decrease of the scavenging activity and an increase in EC_50_ compared to refined olive oil alone. On the contrary, the scavenging activity increased for samples of refined olive oil with 0.1% and 0.5% CBD (Figure 2).

The addition of CBD in sunflower oil, at the three concentrations, led to an increase in scavenging activity compared to sunflower oil alone; in fact, the decrease in EC_50_ values was directly proportional to the concentration of CBD.

The theoretical EC_50_ values for refined olive oil and sunflower oil, used to prepare solutions with the three concentrations of α-tocopherol, were 3.56 g/mL and 12.86 g/mL, respectively.

Refined olive oil with α-tocopherol showed that the addition of 0.01% α-tocopherol decreased the scavenging activity. However, addition of 0.01% and 0.5% α-tocopherol decreased the EC_50_, and increased the scavenging activity (Figure 2).

The addition of α-tocopherol to sunflower oil led to an increase in the scavenging activity vs. sunflower oil alone; the decrease in EC_50_ was directly proportional to the concentration of α-tocopherol.

## 3. Discussion

Two refined oils were chosen since the refining process eliminates most of the antioxidant compounds that are naturally present in these matrices. Moreover, it was possible to compare two oils characterized by a different profile in saturated and unsaturated fatty acids: olive oil is mainly characterized by oleic acid, and sunflower by linoleic acid. In order to correctly and reproducibly compare the parameters related to oxidation in the solutions containing the active ingredients, the same parameters were also determined on the oil samples used as a matrix. Conjugated diene and triene systems, molecules related to a more advanced oxidative state (secondary to peroxides), were not evaluated because CBD absorbs at the same wavelengths, and would thus be an interferent in this analysis. Determination of the content of CBD and α-tocopherol and the recovery of these compounds all showed high values, meaning that both these active compounds dissolved in the oils evaluated.

Sunflower oil initially had a higher peroxide content than refined olive oil, which means that it had a more advanced state of oxidation. Moreover, it was noted that at all concentrations investigated, for both CBD and α-tocopherol, an increase in concentration of the active ingredient determined an increase in the peroxides value. The EU Commission Regulation Ec 702/2007 [24] sets a maximum limit equal to 5 meq O_2_/kg oil, but refers to refined olive oil and not to pharmaceutical grade refined olive oil (Ph Eur degree). For this reason, we referred to the limits established by European Pharmacopoeia (Ph Eur degree) [25] and by Codex Alimentarius [26] giving both a limit equal to 10 meq O_2_/kg oil. The peroxide value of the initial refined olive oil sample, and of the same oil with CBD at 0.01%, 0.1%, and 0.5%, were all below this limit.

The peroxide value of the initial sunflower oil and the same oil with CBD added at 0.01% and 0.1% were lower than the limit fixed by the Codex Alimetarius for refined oils, equal to 10 meq O_2_/kg oil, while the value in the sunflower oil sample with 0.5% CBD was above this limit.

Regarding refined olive oil, both Pharmacopoeia and the Codex Alimentarius specify a limit of 10 meq O_2_/kg of oil, and the peroxide value of refined oil with and without α-tocopherol added at 0.01%, 0.1%, and 0.5% were all below this limit.

The peroxide value of sunflower oil alone or with α-tocopherol at 0.01%, 0.1%, and 0.5% was higher than the limit established by Codex Alimetarius for refined oils, or 10 meq O_2_/kg oil; although this value is above the limit, it was nevertheless used for experimentation since it was preferred to not carry out bleaching treatments, which can remove oxidized polar compounds, in order to not modify the oil native matrix composition.

The values relating to free acidity of sunflower oil and refined olive oil are lower than the limit established by Codex Alimetarius for refined oils of 0.6 mg KOH/g; moreover, the value for refined olive oil is also lower than the limit established by the Pharmacopoeia, namely 0.3 mg KOH/g oil. The values relative to acidity of sunflower oil with CBD or α-tocopherol and refined olive oil with CBD or α-tocopherol at all the concentrations investigated were lower than the limit of 0.6 mg KOH/g established by Codex Alimetarius for refined oils; moreover, the value for refined olive oil with CBD or α-tocopherol was also lower than the Pharmacopoeia limit of 0.3 mg KOH/g oil.

The OSI-time value found for refined olive oil was much higher than that for sunflower oil, which is well-known and understandable given the different characteristic fatty acid compositions of the two oils. In fact, sunflower oil is an oil with a greater intrinsic presence of unsaturated fatty acids, which are much more sensitive to oxidation.

Regarding the oily matrices with CBD added, the results showed a decrease in OSI-time with the addition of the active ingredient, which was increased at higher concentrations. These data agree with what emerged from the determination of peroxides.

Regarding α-tocopherol added to sunflower oil, the increases in OSI-time values were directly proportional to the concentration. As far as refined olive oil is concerned, we recorded a decrease in OSI-time at the two lowest concentrations of α-tocopherol, while at 0.5% α-tocopherol the OSI-time was very similar to the initial value. In fact, according to the literature, the activity of α-tocopherol is slightly antioxidant and sometimes pro-oxidative [27]

The data obtained from the ESR analysis for refined olive oil supplemented with CBD show that this compound may have a slight pro-oxidant effect at the lowest concentration (corresponding to 0.01% of CBD). However, at the other two concentrations investigated, CBD had antioxidant action, causing a decrease in free radicals.

The DPPH^•^ radical scavenging activity assay showed that refined olive oil with CBD at a concentration of 0.01% has a lower scavenging capacity than refined olive oil without additions. Moreover, when increasing the CBD concentration, the DPPH^•^ radical scavenging activity assay showed a marked increase in scavenging capacity, emphasizing its antioxidant power, in agreement with what has been found in the literature, CBD is able to suppress DPPH^•^ [28]; this trend agrees with what was detected by the ESR assay. The results of the ESR test on sunflower oil with CBD show that only 0.5% CBD had an antioxidant effect. These results differ from what emerged from the DPPH^•^ radical scavenging activity assay, where at each concentration tested there was a greater scavenging activity compared to sunflower oil without additions. It is essential to highlight that the EC50 of the DPPH^•^ value considered is not a kinetic parameter and, therefore, does not express antioxidant or anti-radical activity in a lipidic model system during its oxidation, as also indicated by Foti, 2015. In fact, in the DPPH^•^ radical scavenging activity assay, many compounds react rapidly with DPPH^•^ and more slowly with ROO^•^, also according to the solvent used [29].

The results of the ESR test on refined olive oil with α-tocopherol showed a marked antioxidant action of this compound, especially at concentrations of 0.1 and 0.5%. At a concentration of 0.01%, although there was a decrease in the concentration of free radicals, the value was not significantly different from those recorded for refined oil, since it falls within the standard deviation (see Table 3 and Table 4). These results are consistent with the findings found in the DPPH^•^ radical scavenging activity assay, with the only difference being that at the highest concentration (0.5% of α-tocopherol) the scavenging activity does not show a significant difference compared to that of refined olive oil.

The ESR data for sunflower oil with α-tocopherol are consistent with the results of the DPPH^•^ radical scavenging activity assay, except for the sample with 0.01% α-tocopherol, wherein a slight increase in scavenging activity was noted compared to sunflower oil without additions. The data from the DPPH^•^ radical scavenging activity assay showed greater scavenging activity of CBD compared to α-tocopherol.

No correlation was found between the peroxide value and antioxidant capacity, as measured by the DPPH^•^ radical scavenging activity assay. In fact, the peroxide value increased with an increase in the concentration of both CBD and α-tocopherol, while a lower theoretical value of EC_50_ was found for refined oil and sunflower oil with CBD, at all concentrations investigated, and for sunflower oil with α-tocopherol at all concentrations tested. On the other hand, regarding refined oil added with α-tocopherol, the theoretical EC_50_ value remained stable at α-tocopherol concentrations of 0.01% and 0.1%, while it underwent a slight increase at an α-tocopherol concentration of 0.5%.

The lack of a clear and definite correlation between these two assays for some oily matrices has also been reported in previous studies [30,31].

Moreover, the decrease in the induction period (OSI-time) in samples of refined olive oil with α-tocopherol, especially at 0.01% and 0.1%, agrees with previous literature data, according to which analysis of the induction period carried out at 110 ° C does not have a positive correlation with the total tocopherol content [32,33].

## 4. Materials and Methods

### 4.1. Sample Preparation

The antioxidant capacity of cannabidiol (CBD) and α-tocopherol (AT) was evaluated on oily solutions prepared with refined olive oil pharmaceutical grade according European Pharmacopoeia (Ph Eur) and sunflower oil at three different concentrations: 0.01%, 0.1%, and 0.5%. For preparation of these solutions, refined olive oil Ph Eur grade acquired in a pharmacy was used; sunflower oil was from commercial sources; the CBD used was in the form of crystals with 99.8% purity provided by Enecta Srl, Amsterdam (Netherlands), while AT with 97% purity was supplied by Alfa Aesar Thermo Fisher (Erlensbachweg 2, 76870 Kandel, Germania, Germany). In order to obtain complete dissolution of active principles, preparations were placed in an ultrasonic bath, model Branson 2150 (Danbury, CT, USA) for 1 h, divided into four intervals of 15 min, in order to avoid overheating of the solutions. Fifty grams of each solution was prepared and placed in a 100 mL Sovirel bottle. Bottles were stored in the fridge at 4 °C until analysis.

### 4.2. Determination of CBD Content

To determine the actual CBD content and ensure the total dissolution of CBD crystals, solutions were analyzed by liquid chromatography. A total of 100 mg of sample was weighed in a 10 mL flask, solubilized, brought to volume with isopropanol, vortexed for 1 min, and placed in an ultrasonic bath (Branson 2150) for 10 min. Next, the solution was filtered through a 0.45 µm nylon filter. CBD determination was performed following the method proposed by Mandrioli et al. 2019 [34]; 5 µL was injected into an HPLC-UV, Cannabis Analyzer for Potency Prominence-i LC-2030C chromatographic system, equipped with a Nex-Leaf CBX Potency column 150 × 4.6 mm, 2.7 μm (Shimazu, Kyoto, Japan). The eluent mixture eluted in gradient was water with 0.085% phosphoric acid and acetonitrile with 0.085% phosphoric acid. The signal was acquired at 220 nm and processed with LabSolutions version 5.84 software (Shimazu, Kyoto, Japan). Quantification was carried out using a calibration curve constructed with an external standard, injecting solutions of known concentration in the concentration range 0.05–16.70 μg/mL (y = 13065x − 1612.7; R^2^ = 0.9989).

### 4.3. Determination of α-Tocopherol Content

As described above, the oils used to disperse the active ingredients were refined, so that most of the compounds with antioxidant activity such as tocopherols were removed. However, the refining process does not completely remove antioxidant compounds such as tocopherols, and small residual quantities are still present. To evaluate the content of AT in the oily matrix, i.e., natural content plus added content, and to verify the complete dissolution of all active ingredients added, liquid chromatography analysis was performed on prepared oily solutions. The applied method was an internal procedure, briefly described below. An aliquot of 0.5 ± 0.0001 g was solubilized in isopropanol, filtered with 0.45 µm nylon filter, and 20 µL were injected into an RP-HPLC system equipped with a quaternary pump model HP 1260 and fluorimeter detector model HP 1100; the software for data processing was Chemstation for LC3D (Agilent Technologies, Palo Alto, CA, USA). The instrument was also equipped with a column Cosmosil π NAP 150 mm × 4,6 mm Thermo Fisher, 5 µm (Nacalai-Tesque, Kyoto, Japan). The mobile phase was a mixture of: solvent A, methanol–water with 0.2% of H_3_PO_4_ (90:10 *v*/*v*), and solvent B, acetonitrile (100%), eluted in gradient with a flow rate of 1.0 mL/min. The fluorometric detector was set up at an emission wavelength of 295 nm and an excitation wavelength of 330 nm. Quantification was carried out using a calibration curve constructed with the external standard method, injecting solutions of known concentration in the range of 8.34–166.80 μg/mL (y = 42.52x + 57.922; R^2^ = 0.9959).

### 4.4. Peroxide Value

The peroxide value represents a measure of peroxidic compounds, the primary products of lipid oxidation. The determination of peroxide value was carried out according to the NGD C35-1976 method [35], performing an iodometric titration where, following oxide reduction reaction in the presence of starch as an indicator, measures the quantity of peroxide present in the sample expressed as meq of active oxygen per kg of oil.

### 4.5. Free Acidity

The acidity index is defined as mg of KOH needed to neutralize the free acids present in a gram of oil, determined as indicated by the European Pharmacopoeia 9th Edition 2017 [25]. The tested substance, dissolved in a mixture of alcohol and petroleum ether, is titrated with an alkaline hydroxide solution in the presence of phenolphthalein.

### 4.6. Oxidative Stability Index

The oxidative stability index (OSI) was evaluated as the determination of resistance to forced oxidation and was performed using an oxidative stability instrument (OMNION OSI-8 Decatur, IL, USA). A total of 5 g of each sample were weighed in a glass tube and heated to 110 °C in the presence of a continuous flow that reached the end of the path. Through this continuous measurement, the instrument extrapolates the data relative to the induction period from the initial phase of oxidation to that in which it assumes an exponential trend, known as OSI-time [36].

### 4.7. Micro-ESR

The oxidation state of solutions was also investigated using the ESR spectroscopy analysis, a technique that allows evaluation of chemical species with unpaired electrons, such as free radicals. In particular, 1 mL of oily solution was added with 40 µL of 2.5 M *N*-tert-butyl-α-phenylnitrone (PBN, ≥98% VWR-International, Milan, Italy) in ethanol, vortexed for one minute, and readings were taken with microESR STANDARD V 2.0 (Bruker BioSpin GmbH, Rheinstetten, Germany) heating the sample to 110 °C for 240 min, recording the spectra obtained every 20 min. The standard for the calibration curve used was constructed by TEMPO solutions in mineral oil (Sigma-Aldrich, Milan) at concentrations of 1.0, 2.5, 5.0, 10.0, 20.0, 35.0, and 50.0 µM.

### 4.8. DPPH^•^ Radical Scavenging Activity Assay

The extraction of α-tocopherol and CBD was performed following the procedure reported by Ninfali et al. 2001 [37]. A total of 5 g of oily sample were weighed and 5 mL of a mixture of methanol:water 80:20 *v*/*v* was added. It was vortexed for two minutes, and the sample was centrifuged at 2500 rpm for 15 min. The supernatant was removed and placed in a 10 mL flask. The extraction phase was repeated with 4.5 mL of the mixture of methanol:water 80:20 *v*/*v* and, finally, was brought to volume in a 10 mL flask, with the same mixture.

The antioxidant capacity was investigated by applying the DPPH^•^ radical scavenging activity assay on different extracts. Total of 11.9 mg of DPPH^•^ supplied by Thermo Fisher (Kandel, GmbH, Germany) was weighed in a 50 mL volumetric flask, and brought to volume with methanol, obtaining a solution with a concentration of 0.604 mM. This solution was diluted by taking 5 mL and bringing it to volume, in a volumetric flask, with methanol to 50 mL, obtaining a solution with a concentration equal to 0.0604 mM or 60.4 μM.

The spectrophotometry analysis was performed against methanol at 515 nm using a Jasco dual beam spectrophotometer model V-550 UV-VIS, with the possibility of reading each nanometer. Quartz cuvettes with an optical path of 10 mm, supplied by Lightpath Optical (Cranborne, Churchill, Axminster EX13 7LZ, UK), were used. A total of 2.9 mL of 60.4 μM DPPH^•^ solution was placed in the cuvette and read; subsequently, 100 μL of methanol:water 80:20 *v*/*v* was added in this cuvette, thus obtaining a value relative to blank (A_0_), and the absorbance was read. Next, in order to have the absorbance value of the control sample (A_j_), 2.9 mL of methanol was placed in the cuvette and 0.1 mL of the extract (sample) was added and the absorbance was read [38]. For each sample, five different concentrations in the range between 5 mg/mL and 50 mg/mL were analyzed at 515 nm, prepared by placing them in 2.9 mL of 60.4 μM DPPH^•^ solution. The solutions were mixed thoroughly and protected from light at room temperature for 30 min, and radical scavenging was estimated by determining the loss of absorbance at 515 nm [39]. The solutions were kept in the dark for 30 min; this time was chosen according to the literature [40,41,42] and having verified that with it only the oils with CBD at 0.5% reached the plateau of the absorbance value. As all the other solutions reached the plateau within 24 h, 30 min appeared to be the best compromise to compare the results of all DPPH^•^ assays.

The experimental scavenging capacity (ESC) value was calculated according to the following formula [39]:ESC = (1 − [A_i_ − A_j_]/A_0_) × 100,(2) where: A_i_ = absorbance of the sample; A_0_ = absorbance of the blank; A_j_ = absorbance of the control sample.

The effective concentration EC_50_ of DPPH^•^ value was determined for each sample. EC_50_ of DPPH^•^ was defined as the efficient concentration required to decrease the initial DPPH^•^ concentration by 50% [43]. The EC_50_ value was extrapolated from the curve obtained by plotting the ESC value against the five concentrations in the range between 5 mg/mL and 50 mg/mL for each sample.

### 4.9. Statistical Analysis

Samples were analyzed in duplicate and the results are shown as mean ± standard deviation and coefficient of variation (RSD%).

The data presented in Table 1, Table 2, Table 3 and Table 4 were statistically analyzed by applying Anova, Fisher’s test LSD, *p* < 0.05 using the software XLSTAT Addinsoft (2018.XLSTAT statistical and data analysis solutions. Paris, France. https://www.XLSTAT.com) version 2018.1.1.

## 5. Conclusions

For the two model systems of edible oils considered, which differed in terms of fatty acid composition and initial oxidative state, several interesting aspects were found. In particular, an increase in the peroxide value, perhaps linked with oxygenation because of the preparation/mixing of the oily solutions, was observed; furthermore, a certain additional pro-oxidant effect of CBD, confirmed by measuring the oxidation parameters, appeared evident. This pro-oxidant activity was also confirmed by the reduction of the OSI-time analysis, passing from 23.58 ± 0.32 (ROO) to 17.28 ± 0.18 h (ROO with 0.5% CBD) and from 4.93 ± 0.04 (SO) to 3.98 ± 0.04 h (SO with 0.5% CBD). In terms of free radicals, the ESR assay did not show a relevant decrease in their concentration because of the addiction of CBD, with corresponding values of 76.94 ± 9.04 μM (for ROO) and 72.25 ± 4.13 μM (ROO with 0.5% CBD), and extremely similar for the model system made by the pure sunflower oil (SO, 17.91 ± 0.95 μM) and with the addition of 0.5% CBD (16.84 ± 0.25 μM). On the contrary, the DPPH^•^ assay showed a higher scavenging capacity for CBD than for α-tocopherol, likely because of the presence of two hydroxyl groups in the CBD molecule. The addition of 0.1% and 0.5% α-tocopherol, even in the presence of an increase in the number of peroxides, exerted higher antioxidant activity in the more unsaturated system (SO), as shown by the forced oxidation test. In fact, OSI-time increased from 4.15 ± 0.07 to 6.28 ± 0.11 h for sunflower oil with 0.5% α-tocopherol, while the OSI-time of the refined olive oil substantially did not change (form 28.35 ± 0.28 to 28.23 ± 0.81 h). α-Tocopherol, at a concentration of 0.01%, did not have any protective effect, in either of the model systems. It is interesting to notice that the measure of the resistance at the forced oxidation (OSI-time analysis) can be interpreted in a certain coherence with that of the free radicals (by ESR assay), while the data of DPPH^•^ assay showed a different trend. In other words, the higher scavenging activity of CBD, measured by DPPH^•^, does not seem to be related with a greater oxidative stability of the model system when added to it. This can be due to the different reaction kinetics of the species considered with respect to the DPPH^•^ and of the peroxide radical forms ROO^•^ present in the matrix. As highlighted in literature, the use of this (DPPH^•^), or similar scavenging test, to compare the antioxidant activities of different molecules is questionable. To provide an exhaustive framework, it will necessary to thoroughly investigate the conditions and concentration at which CBD shows an antioxidant or a pro-oxidant effect, knowing that the use of a single test could be misleading. Regarding the two model oily solutions considered herein, more or less unsaturated, it is essential to note that the CBD did not show a protective antioxidant action when added up to 0.5%. This evidence is useful to establish the shelf-life of oily solutions, very common and diffuse in the market, containing CBD and to also correctly formulate oily foods, medicines, or supplements containing CBD. Concerning shelf-life, it will be necessary to study the information given by the ESR-forced oxidation assay, such as coupling constants, type of radicals, and interaction among PBN, lipid radicals, and added molecules, to see if this test could be effectively and reliably use as predictive.

## Figures and Tables

**Figure 1 molecules-24-03485-f001:**
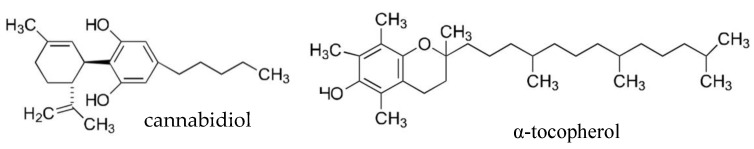
Chemical structures of cannabidiol and α-tocopherol.

**Figure 2 molecules-24-03485-f002:**
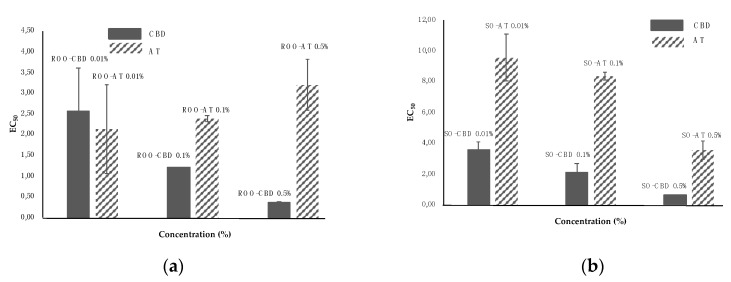
Differences in EC_50_ values of DPPH^•^ obtained by adding CBD or α-tocopherol to refined olive oil (ROO) on the left (**a**), and to samples obtained by adding CBD and α-tocopherol to sunflower oil (SO) on the right (**b**). A lower value of EC_50_ corresponds to higher scavenging activity.

**Table 1 molecules-24-03485-t001:** Results of peroxide value, free acidity, and OSI-time for refined olive oil (ROO), and sunflower oil (SO) with cannabidiol (CBD) added. Data are presented as mean ± standard deviation and coefficient of variation (RSD%).

Sample	Peroxide Value (meq O_2_/Kg of Oil)	RSD %	Free Acidity (mg KOH/g of Oil)	RSD%	OSI-Time (Hours)	RSD%
ROO	1.93 ± 0.09 ^c^	4.66	0.15 ± 0.02 ^b^	12.86	23.58 ± 0.32 ^a^	1.35
ROO-CBD 0.01%	2.47 ± 0.04 ^b^	1.59	0.18±0.02 ^b^	11.70	23.33 ± 0.18 ^a^	0.76
ROO-CBD 0.1%	2.98 ± 0.14 ^a^	4.66	0.18±0.02 ^b^	10.88	21.90 ± 0.07 ^b^	0.32
ROO-CBD 0.5%	2.99 ± 0.14 ^a^	4.68	0.27±0.02 ^a^	7.30	17.28 ± 0.18 ^c^	1.02
SO	5.00 ± 0.02 ^X^	3.00	0.11±0.00 ^X^	0.00	4.93 ± 0.04 ^X^	0.72
SO-CBD 0.01%	6.30 ± 0.01 ^Y^	0.13	0.08±0.00 ^Y^	0.00	4.80 ± 0.21 ^Y^	4.42
SO-CBD 0.1%	6.04 ± 0.29 ^Y^	4.76	0.06±0.00 ^Z^	0.07	4.90 ± 0.00 ^Y^	0.00
SO-CBD 0.5%	14.73 ± 0.24 ^Z^	1.60	0.06±0.00 ^Z^	0.07	3.98 ± 0.04 ^Z^	0.89

ROO = refined olive oil. ROO-CBD = refined olive oil with CBD, the percentage indicates the CBD/oil ratio. SO = sunflower oil. SO-CBD = sunflower oil with CBD, the percentage indicates the CBD/oil ratio. Different letters indicate statistically significant differences (Anova, Fisher’s test LSD, *p* < 0.05).

**Table 2 molecules-24-03485-t002:** Results of peroxide value, free acidity, and OSI-time for refined olive oil (ROO) and sunflower oil (SO) with α-tocopherol (AT) added. Data are presented as mean ± standard deviation and coefficient of variation (RSD%).

Sample	Peroxide Value (meqO_2_/Kg of Oil)	RSD %	Free Acidity (mg KOH/g of Oil)	RSD %	OSI-Time (Hours)	RSD %
ROO	0.15 ± 0.00 ^c^	0.08	0.17 ± 0.00 ^b^	0.07	28.35 ± 0.28 ^a^	1.00
ROO-AT 0.01%	0.52 ± 0.03 ^b^	6.65	0.17 ± 0.00 ^b^	0.00	25.30 ± 0.42 ^b^	1.68
ROO-AT 0.1%	0.52 ± 0.04 ^b^	6.78	0.21 ± 0.02 ^a^	9.43	24.88 ± 0.18 ^b^	0.71
ROO-AT 0.5%	0.62 ± 0.04 ^a^	5.67	0.21 ± 0.02 ^a^	9.43	28.23 ± 0.81 ^a^	2.88
SO	12.31 ± 0.50 ^X^	4.03	0.20 ± 0.00 ^X^	0.07	4.15 ± 0.07 ^x^	1.70
SO-AT 0.01%	13.50 ± 0.09 ^X,Y^	0.64	0.21 ± 0.02 ^X^	9.15	4.20 ± 0.07 ^x^	1.68
SO-AT 0.1%	13.43 ± 0.70 ^X,Y^	5.18	0.17 ± 0.00 ^Y^	0.14	5.28 ± 0.04 ^y^	0.67
SO-AT 0.5%	15.08 ± 1.16 ^Y^	7.68	0.17 ± 0.00 ^Y^	0.00	6.28 ± 0.11 ^z^	1.69

ROO = refined olive oil. ROO-AT = refined olive oil with α-tocopherol, the percentage indicates the α-tocopherol/oil ratio.SO = sunflower oil. SO-AT = sunflower oil with α-tocopherol, the percentage indicates the α-tocopherol/oil ratio. Different letters indicate statistically significant differences (Anova, Fisher’s test LSD, *p* < 0.05).

**Table 3 molecules-24-03485-t003:** Results of electron spin resonance (ESR)-forced oxidation assay after 20 min and 240 min for refined olive oil (ROO) and sunflower oil (SO) with cannabidiol (CBD). Data are presented as mean ± standard deviation and coefficient of variation (RSD%).

Sample	Concentration of Free Radicals after 20 min (µM)	RSD %	Concentration of Free Radicals after 240 min (µM)	RSD %
ROO	5.70 ± 0.25 ^b^	4.47	76.94 ± 9.04 ^a^	11.26
ROO-CBD 0.01%	10.39 ± 0.57 ^a^	5.44	83.85 ± 5.45 ^a^	6.50
ROO-CBD 0.1%	9.27 ± 0.31 ^a^	3.36	70.98 ± 5.62 ^a^	7.92
ROO-CBD 0.5%	6.50 ± 0.25 ^b^	3.81	72.25 ± 4.13 ^a^	5.72
SO	8.35 ± 0.22 ^X^	2.66	17.91 ± 0.95 ^X^	5.29
SO-CBD 0.01%	9.87 ± 0.00 ^Y^	0.00	20.82 ± 0.53 ^X^	2.57
SO-CBD 0.1%	9.00 ± 0.53 ^X,Y^	5.86	25.37 ± 2.27 ^Y^	8.96
SO-CBD 0.5%	5.41 ± 0.24 ^Z^	4.51	16.84 ± 0.25 ^Z^	1.47

ROO = refined olive oil. ROO-CBD = refined olive oil with CBD, the percentage indicates the CBD/oil ratio. SO = sunflower oil. SO-CBD = sunflower oil with CBD, the percentage indicates the CBD/oil ratio. Different letters indicate statistically significant differences (Anova, Fisher’s test LSD, *p* < 0.05).

**Table 4 molecules-24-03485-t004:** Results of ESR-forced oxidation assay after 20 min and 240 min for refined olive oil (ROO) and sunflower oil (SO) with α-tocopherol (AT) added. Data are presented as mean ± standard deviation and coefficient of variation (RSD %).

Sample	Concentration of Free Radicals after 20 min (µM)	RSD %	Concentration of Free Rad icals after 240 min (µM)	RSD %
ROO	3.10 ± 0.15 ^a^	4.72	83.33 ± 4.56 ^a^	5.48
ROO-AT 0.01%	3.19 ± 0.03 ^a^	0.89	81.22 ± 6.04 ^a^	7.43
ROO-AT 0.1%	1.27 ± 0.14 ^a^	11.14	61.41 ± 5.82 ^b^	9.48
ROO-AT 0.5%	0.55 ± 0.03 ^b^	5.81	11.23 ± 0.28^c^	2.52
SO	5.90 ± 0.27 ^X^	4.55	19.21 ± 1.39 ^X^	7.22
SO-AT 0.01%	6.66 ± 0.23 ^Y^	3.43	22.61 ± 0.23 ^Y^	1.00
SO-AT 0.1%	3.35 ± 0.28 ^Z^	8.24	13.25 ± 0.74 ^Z^	5.59
SO-AT 0.5%	2.08 ± 0.20 ^W^	9.47	6.90 ± 0.53 ^W^	7.64

ROO = refined olive oil. ROO-AT = refined olive oil with α-tocopherol, the percentage indicates the α-tocopherol/oil ratio. SO = sunflower oil. SO-AT = sunflower oil with α-tocopherol, the percentage indicates the α-tocopherol/oil ratio. Different letters indicate statistically significant differences (Anova, Fisher’s test LSD, *p* < 0.05).

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
