# Peer review of "Preliminary Study: Comparison of Antioxidant Activity of Cannabidiol (CBD) and α-Tocopherol Added to Refined Olive and Sunflower Oils"

_molecules, 2019, doi:10.3390/molecules24193485_

Round 1

Reviewer 1 Report

The manuscript presents a very interesting research topic, supported by the obtained results. The only observation that I have, is to make some statistics for the results presented in Tables 1 and 2. Also, please consider changing the colors for  Figure 2, for a better view in the printed article.

After these small observations, I would recommend the publication, with the agreement of the Editorial Board. 

Author Response

Point 1: The only observation that I have, is to make some statistics for the results presented in Tables 1 and 2. Also, please consider changing the colors for Figure 2, for a better view in the printed article.

Response 1: We thank the reviewer for the observation, the suggested changes were made in the revised version. In fact, some statistics for the results presented in Table 1 and 2 were applied, in particular ANOVA, Fisher’s test LSD. Moreover, the colors of Figure 2 were changed as suggested.

Reviewer 2 Report

The authors of this paper present an interesting preliminary study on the comparison of antioxidant activity of cannabidiol (CBD) and α-tocopherol added to refined olive and sunflower oils. Nevertheless, this work might be improved taking into account the following suggestions:

-              Generally an extensive English language correction should be performed, as it is not clear at some points and the writing is sometimes difficult to follow.

For example please give the definition of the free radical DPPH∙ as 2,2-diphenyl-1-picrylhydrazyl  and correct it as DPPH∙  while the ABTS need to be corrected as ABTS•+  in line 61 and in all manuscript. Line 95-96 The mean value obtained for the recovery of CBD in refined olive oil and in sunflower oil was 110.15%, and thus it can be considered completely solubilized ??? The starting concentration in ppm? Please explain further. Line 258 & 260 Ph Eur, the definition of the abbreviation is needed. Also, European Pharmacopeadia and the Regulation Ec 702/2007 are two different things please define the connection.

-              Abstract need to rewrite up to 200 words (according to the journal instructions) and should show quantitative relevant findings of the study. Same for the Conclusions.

- Line 63:  The addition of the following reference is been proposed.

“Skenderidis, P.; Lampakis, D.; Giavasis, I.; Leontopoulos, S.; Petrotos, K.; Hadjichristodoulou, C.; Tsakalof, A. Chemical Properties, Fatty-Acid Composition, and Antioxidant Activity of Goji Berry (Lycium barbarum L. and Lycium chinense Mill.) Fruits. Antioxidants 20198, 60.“

-              Please add error bars in Figure 2.

-              Results presented generally in somehow confused. For example, in lines 106-120 is required an explanation about the differences in initial peroxide values and why the initial starting samples have different peroxide values? Why you didn't use the same initial samples?

-              The discussion is poorly written and should be revised and re-written well and compare your results with other papers.

- In statistical analysis please report the statistical software you have used in order to make the analysis.

Author Response

Point 1:Generally an extensive English language correction should be performed, as it is not clear at some points and the writing is sometimes difficult to follow. 

For example please give the definition of the free radical DPPH∙ as 2,2-diphenyl-1-picrylhydrazyl  and correct it as DPPH∙  while the ABTS need to be corrected as ABTS•+  in line 61 and in all manuscript. Line 95-96 The mean value obtained for the recovery of CBD in refined olive oil and in sunflower oil was 110.15%, and thus it can be considered completely solubilized ??? The starting concentration in ppm? Please explain further. Line 258 & 260 Ph Eur, the definition of the abbreviation is needed. Also, European Pharmacopeadia and the Regulation Ec 702/2007 are two different things please define the connection.

Response 1: We would like to thank the reviewer for highlighting these points and typos. We gave the definition of DPPH• as 2,2-diphenyl-1-picrylhydrazyl and we corrected DPPH in DPPH• and ABTS in ABTS•+. Moreover, in the revised manuscript we explain further the paragraph about the recovery of CBD.

The definition of the abbreviation of Ph Eur was added in the revised version. Moreover, we changed the sentence in order to better understand that the Regulation EC 702/2007 refers to refined olive oil and not to pharmaceutical grade refined olive oil.

Point 2: Abstract need to rewrite up to 200 words (according to the journal instructions) and should show quantitative relevant findings of the study. Same for the Conclusions.

Response 2:We thank the reviewer for the suggestion, the abstract and the conclusions were revised and some results were presented.

Point 3: Line 63:  The addition of the following reference is been proposed.

“Skenderidis, P.; Lampakis, D.; Giavasis, I.; Leontopoulos, S.; Petrotos, K.; Hadjichristodoulou, C.; Tsakalof, A. Chemical Properties, Fatty-Acid Composition, and Antioxidant Activity of Goji Berry (Lycium barbarum L. and Lycium chinense Mill.) Fruits. Antioxidants 2019, 8, 60.“

Response 3: We thank the reviewer for this suggestion, we added the reference proposed.

Point 4: Please add error bars in Figure 2.

Response 4: The error bars were added in the revised version.

Point 5: Results presented generally in somehow confused. For example, in lines 106-120 is required an explanation about the differences in initial peroxide values and why the initial starting samples have different peroxide values? Why you didn't use the same initial samples?

Response 5: We would like to thank the reviewer for this highlighting, we added the explanation requested about the differences in initial peroxide value.

Point6: The discussion is poorly written and should be revised and re-written well and compare your results with other papers.

Response 6: We thank the reviewer for the suggestions, we revised the discussion and we added some references. Moreover, a native English revised the language in order to better present the results and discussion sections.

Point 7: In statistical analysis please report the statistical software you have used in order to make the analysis.

Response 7: We would like to thank the reviewer; the name of the statistical software was added in the revised version.

Reviewer 3 Report

This is a study on assessment of antioxidant capacity of CBD in comparison with a-tocopherol under different conditions. The authors measured several parameters by different methods, but, as stated in the Conclusion, the results are not straightforward and some results are hard to understand. As the authors state (lines 469-470), it is necessary to clarify some issues in the future study. Following points should be addressed.

1.     Tables. It is not easy to evaluate the data in the Tables because of many abbreviations. It may be advised to show what the abbreviations mean at the footnote of Tables.                                   2.     ESR study. It is advised to show the ESR spectrum and coupling constants. Any information which kind of radicals were trapped by a spin trap PBN? The peroxyl radical spin adduct of PBN is known to be unstable. Please state how the radicals were quantified. How the temperature 110 degree chosen? 

3.     DPPH assay. The antioxidant capacity obtained depends on the experimental conditions such as concentration and time. In the present experiments, the solution was kept for 30 min before reading. How the time 30 min determined? Does the time affect the results?

Author Response

Point 1: Tables. It is not easy to evaluate the data in the Tables because of many abbreviations. It may be advised to show what the abbreviations mean at the footnote of Tables.   

Response 1: We would like to thank the reviewer for this suggestion, we added the footnotes in order to better understand the abbreviations reported in the tables.

Point 2: ESR study. It is advised to show the ESR spectrum and coupling constants. Any information which kind of radicals were trapped by a spin trap PBN? The peroxyl radical spin adduct of PBN is known to be unstable. Please state how the radicals were quantified. How the temperature 110 degree chosen?

Response 2: We would like to thank the reviewer for these suggestions. We added some information about the PBN spin trap technique but, since it is a preliminary study, we have not yet investigated the coupling constants nor identified the radicals involved, it is certainly one of the future developments of this work necessary to better understand the results achieved. To date, we have only quantified the free radicals without identified them. For the quantification of free radicals, the instrument used allows an automatic quantification thanks to a calibration based on a calibration curve constructed by TEMPO solutions, as reported in the section 4.7. Moreover, we added why we chose 110°C.

Point 3: DPPH assay. The antioxidant capacity obtained depends on the experimental conditions such as concentration and time. In the present experiments, the solution was kept for 30 min before reading. How the time 30 min determined? Does the time affect the results?

Response 3: We would like to thank the reviewer for these questions. In the revised version we added how 30 minutes were determined. As regarding the incubation time, by modifying the time also the percentage of inhibition were modified, for this reason we calculated the EC50as a theoretical value. However, we chose an incubation time of 30 minutes as it is often reported in the literature and, in addition, the oils with CBD at 0.5% reached the absorbance plateau after 30 minutes, in this way we applied the same conditions of incubation time to all samples.